# Multi-Use of the Sea as a Sustainable Development Instrument in Five EU Sea Basins

Joanna Przedrzymirska [1,*], Jacek Zaucha [2,3], Helena Calado [4], Ivana Lukic [5], Martina Bocci [6], Emiliano Ramieri [7], Mario Cana Varona [8], Andrea Barbanti [9], Daniel Depellegrin [9,10], Marta de Sousa Vergílio [11], Angela Schultz-Zehden [5], Vincent Onyango [12], Eva Papaioannou [12], Bela H. Buck [13,14], Gesche Krause [13], Maximilian Felix Schupp [12,13], Rianne Läkamp [15], Kazimierz Szefler [2], Monika Michałek [2], Mairi Maniopoulou [16], Vassiliki Vassilopoulou [16], Zacharoula Kyriazi [16,17], Krystyna Gawlikowska-Hueckel [3], Stanisław Szultka [18], Christian Orobello [18,19], Kira Gee [20], Bruce Buchanan [21] and Marija Lazić [22]



1 The Institute of Oceanology of the Polish Academy of Sciences, 81-712 Sopot, Poland
2 Maritime Institute in Gdańsk, Gdynia Maritime University, 80-830 Gdańsk, Poland; j.zaucha@instytut-rozwoju.org (J.Z.); kszefler@im.umg.edu.pl (K.S.); mmichalek@im.umg.edu.pl (M.M.)
3 Institute for Development in Sopot, 81-706 Sopot, Poland; k.gawlikowska-hueckel@instytut-rozwoju.org
4 Faculty of Science and Technology, Marine Environmental Sciences Centre, UAc/FCT-MARE, University of the Azores, 9501-801 Ponta Delgada, Portugal; helena.mg.calado@uac.pt
5 SUBMARINER Network, 10827 Berlin, Germany; il@sustainable-projects.eu (I.L.); asz@submariner-network.eu or asz@sustainable-projects.eu (A.S.-Z.)
6 t-ELIKA SRL, 30133 Venice, Italy; martina.bocci@unive.it
7 Thetis, 30122 Venice, Italy; emiliano.ramieri@thetis.it
8 Biology Department, School of Science and Technology, UAc/FCT-MARE, University of the Azores, 9501-801 Ponta Delgada, Portugal; mario.cana@grida.no
9 CNR-ISMAR, 30122 Venice, Italy; andrea.barbanti@ve.ismar.cnr.it (A.B.); daniel.depellegrin@udg.edu (D.D.)
10 Ocean & Human Health Chair, Institute of Aquatic Ecology, University of Girona, 17003 Girona, Spain
11 CIBIO – Research Center in Biodiversity and Genetic Resources/InBIO – Associate Laboratory, University of the Azores, 9501-801 Ponta Delgada, Portugal; marta.hs.vergilio@uac.pt
12 Department of Architecture and Urban Planning, School of Social Sciences, University of Dundee, Dundee DD1 4HN, UK; v.onyango@dundee.ac.uk (V.O.); epapaioannou@geomar.de (E.P.); maximilian.felix.schupp@awi.de (M.F.S.)
13 Marine Aquaculture, Biosciences, Alfred Wegener Institute, Helmholtz Centre for Polar and Marine Research, 27570 Bremerhaven, Germany; Bela.H.Buck@awi.de (B.H.B.); Gesche.Krause@awi.de (G.K.)
14 Applied Marine Biology and Aquaculture, University of Applied Sciences, 27568 Bremerhaven, Germany
15 Ecorys, 1040 Brussels, Belgium; rianne.lakamp@rws.nl
16 Hellenic Centre for Marine Research, 19013 Attiki, Greece; maniopoulou@hcmr.gr (M.M.); celia@hcmr.gr (V.V.); zkyriazi@ciimar.up.pt (Z.K.)
17 Interdisciplinary Centre of Marine and Environmental Research, University of Porto, 4450-208 Matosinhos, Portugal
18 Department of Macroeconomics, Faculty of Economics, University of Gdańsk, 80-309 Gdańsk, Poland; stanislaw.szultka@ug.edu.pl (S.S.); christian.orobello@ug.edu.pl (C.O.)
19 Managing and Leading in Business, Southern New Hampshire University, Manchester, NH 03106, USA
20 Helmholtz-Zentrum Hereon, 21502 Geesthacht, Germany; Kira.Gee@hereon.de
21 Marine Scotland, Edinburgh EH6 6QQ, UK; Bruce.buchanan@gov.scot
22 Institute of Transportation, Nemanjina 6/IV, 11000 Belgrade, Serbia; marija.lazic@sicip.co.rs
* Correspondence: jprzedrzymirska@iopan.pl; Tel.: +48-609-005-055

**Abstract:** This paper examines the concept of maritime multi-use as a territorial/SPATIAL governance instrument for the enhancement of sustainable development in five EU sea basins. Multi-use (MU) is expected to enhance the productivity of blue economy sectors, as well as deliver additional socio-economic benefits related to the environmental and social dimensions of sustainable development. The paper provides a definition of maritime multi-use and identifies the multi-uses with the highest potential in EU sea basins. In each sea basin, multi-use plays a different role as concerns sustainable development. For the Eastern Baltic Sea, the Mediterranean Sea and the Black Sea, the MU focus should remain on the environmental pillar of sustainable development. In the North Sea, North Atlantic and Western Baltic Sea, addressing social sustainability seems a key precondition for success of MU in enhancement of sustainable spatial development at sea. Moreover, it has been

suggested to introduce MU key global strategies such as SDGs or Macroregional strategies and action plans and to supplement maritime spatial planning with sectoral incentives and educational efforts as key vehicles supporting MU. The paper concludes by identifying aspects which, in order to inform maritime spatial planning and maritime governance regarding a more conscious application of the aforementioned concept, require further investigation. Key tasks are related to: more profound evaluation of performance of policies supporting MUs, researching the impact of MU on societal goals and on the MU costs and benefits, including external ones, and finally identifying the impact of MU on the development of various sectors and regions on land.

**Keywords:** multi-use; blue growth; marine space; marine policy

## 1. Introduction

Conserving and sustainably uing oceans, seas, and marine resources constitutes one of the key Sustainable Development Goals (SDGs) of the United Nations Agenda 2030 [1]. The key issue in this context is an amplification of human utilization of marine space and the latter's more extensive exploitation for economic purposes [2]. As a result, marine sea space is becoming a scarce resource, requiring careful management, and can no longer be perceived as infinite and abundant [3]. Yet, as pointed out by Medeiros, the SGDs lack policy support for spatial planning [4], a problem since maritime spatial planning (MSP) is considered a promising governance vehicle able to ensure smart trade-offs between various components of sustainable spatial development [5]. In these circumstances, concepts such as multi-use (MU), co-location or co-existence have recently attracted the attention of researchers and decision makers [6–20]. The paper has two aims. The first is to propose the options and patterns for multi-use territorialisation in line with the characteristics of EU sea basins. The second is to identify the research gaps that need to be overcome if the concept of MU is to become an agent (instrument of territorial governance) for enhancement of such development in the EU seas and oceans. The added value of the research is a systematic positioning of MU against the concept of sustainable development—in particular its three dimensions. Social marine sustainability has been conceptualised only recently [21], and therefore this paper elaborates on a link between MU and redefined marine sustainability as normative milestones of maritime spatial development. The paper is composed of four parts. In the first section, key concepts are introduced and analysed. In the second part, the research methodology is presented. In the third part, the results and discussion serve to identify the state-of-the-art of MU deployment and offer recommendations on policy support for MU as a part of sustainable development. The concluding section provides an agenda for further research.

## 2. Sustainable Development and Multi-Use

### 2.1. Key Pillars of Maritime Sustainable Development

The sustainable development of seas and oceans rests on three traditional pillars that form the backbone of this concept (i.e., economic development, environmental responsibility, and social progress) [22]. In the EU marine context, economic development is attributed to the concept of blue growth [23] defined as 'smart, sustainable and inclusive economic and employment growth from the oceans, seas and coasts' [24]. Formulated at the Rio + 20 Earth Summit in 2012, the concept came as a response to the need to eradicate poverty [25]; nowadays, it represents one of the key EU economic strategies [26]. The *Marine* Strategy *Framework Directive* (MFD), which aims to achieve a Good Environmental Status (GES) of the EU's seas and oceans [27], provides the crucial element of environmental responsibility. Several scholars underline mismatches, tensions, and discrepancies between these two approaches (i.e., MFD and blue growth) [28,29]. The social sustainability of maritime spatial development has not been addressed directly and only partially indirectly through the EU MSP Directive [6]), which encourages the participation of various stakeholders in the MSP

process. This directive also underlines the importance of co-existence. Thus, while social sustainability at sea has not been conceptualised in formal documents of EU legislation, its essence was recently outlined by MSP researchers [26]. They have identified Recognition, Representation, and Distribution as interdependent and interwoven building blocks that, together, contribute towards conceiving social sustainability as a pillar of sustainability at sea. This first element means recognition of (respect in relation to) the diversity of group identity (and related socio-cultural rights, needs, livelihoods, lifestyles, and knowledge). The second one concerns inclusion in and exclusion from the decision-making process. The third one covers the distribution of goods and bads as a result of the governance process.

The situation outlined above makes clear that maritime sustainable development encapsulates multiple expectations and demands, such as GDP growth and employment, poverty alleviation, ecological sensitivity, as well as respect for spatial justice and diversity. Any trade-offs between those dimensions require deliberation through a process of public choice [30]. Multi–use may help juxtapose the aforesaid dimensions of sustainable maritime development. Thus, it should be considered in various marine governance processes, including the MSP, as the actions pursued in agreement with the aims of these dimensions will ultimately convert into various spatial demands and arrangements.

### 2.2. Multi–Use as at Sea

Multi-use (MU) is one of several terms describing the situation of at least two marine sectors or activities being together. Here, the term being together refers to either spatial proximity, overlap or concurrence, or economic interaction. Such a situation can be described also as multiple-use, co-use, coexistence, interdependencies and co-location. Several researchers use some of these terms interchangeably (e.g., [11,31,32]). However, in this paper MU is narrowed to multi-functional and symbiotic combinations with clear economic or social or environmental interplay [33]. The reason for this choice is to leave aside co-existence that will not contribute at least to one of the sustainable development pillars or has nothing to do with territorial governance because its incidental character. Therefore, in this paper, MU is understood as the intentional joint clustering of two or more uses for the purpose of using the same infrastructure and/or using resources in close geographic proximity. Thus, in terms of terminology, MU shares a locational element with co-location (proximity), the excess of revenues/benefits over costs with co-existence (mutual interactions between uses), and resource sharing with co-use (using the same resource). What makes it distinct is the intentionality behind MU and the idea of actively bringing elements together. An example of MU can be a wind farm designed in such a way that it also allows the harvesting of tidal energy (lower costs and extra revenues by using the same infrastructure) [13]. Similarly, the combination of a fishery with environmental protection or tourism can be considered MU if done intentionally and benefiting both activities. Thus, the condition sine qua non for the occurrence of MU is the intentional creation of lower costs and/or extra revenues/benefits from jointly using the same ocean resource (e.g., ocean space, water, fish. etc.) or cross-sectoral operational synergies (joint use or installations, vessels, human resources etc.) that trigger intentional decisions and interactions.

Lower coasts contribute to the economic pillar of sustainable development. However, in many cases, MUs offer various socio-environmental benefits (cf. [8,34]) that cannot be easily monetised within the market process. For instance, MUs may facilitate the survival of sectors with limited market power (driven out from the sea by stronger counterparts), such as an artisanal fishery. Furthermore, a number of environmental benefits can arise from shared use of infrastructure and resources (e.g., increase of popular support for conservation of marine ecosystem due to combination of protection with high quality tourism in order to show the hidden beauty of the marine protected areas to the visitors/divers). Thus, a feature specific to MU is its focus on a more efficient use of resources or the creation of other socio-economic benefits through intentional co-use.

According to Przedrzymirska et al., [35,36] there are two main forces that may enhance MU development: market and policies, both of which are amplified by research and development (R&D). Market forces refer to the economic benefits gained from the combination of several sea uses in terms of lower costs or extra revenue streams for the business sector. This accounts for e.g., the spontaneous emergence of pescatourism where the same resource (fishing boat) is shared by the fishing and tourism sector. Policy drivers are primarily concerned with the goal of attaining/maintaining good environmental status or supporting the existence of sunset industries important for cultural reasons (identity, emotional bond—cf. [37]). Policy drivers can also aim to facilitate greater social acceptance of space-intensive sectors (e.g., aquaculture and renewable energy), e.g., by requiring such sectors to share sea space [32] and by promoting co-location as a way of using ocean space sparingly so space is left for future generations. If MU as a policy has the additional goal of providing extra socio-economic benefits for the parties not involved in MUs (external benefits), policies should provide incentives toward MU. An example may be the incentivised insurance costs for multi-use wind farm structures which produce energy and contribute to higher water quality (plant-based or crustacean aquaculture). Also, MSP can enhance MU in maritime spatial plans (e.g., by preferences to MU arrangements when allocating marine space).

## 3. Materials and Methods

The data and information used in this paper were collected in the years 2017–2018, as part of the Horizon 2020 project named MUSES. The detailed description of the research methodology is included in the project's materials [38]. For the purpose of this paper, in order to identify the potential for various MUs in five EU sea basins, only a small amount of data has been used. The detailed results of MUSES have been presented in several scientific papers [8,34,39,40].

The research that forms the basis for this paper began by analysing MUs at the national level (all 23 EU coastal countries); thereafter, the results were aggregated and analysed at the sea basin level. All five EU sea basins were analysed: the North-East Atlantic (referred hereafter as Atlantic), the North Sea, the Baltic Sea, the Mediterranean Sea, and the Black Sea [35]. If a country belongs to two or more sea basins (e.g., France), analyses related to sea use were conducted separately for each sea basin, while the policy-relevant analyses were done at the national level.

Research first identified MUs and MU-relevant policies during the course of a desktop study covering various transnational EU projects, legal acts, policy documents, and reports both in English and in the respective national languages. Internet sites of various stakeholders were also analysed. This resulted in the identification of existing MUs and the uses constituting them, the shared resources for each case, the location of the MUs, their maturity level (pilot phase or full deployment as well as technology readiness level), the stakeholders involved (both public and private), development patterns (MUs initiated by one user or by two or more users together), and, finally, the legal basis and policy support for the MUs. In some cases, the advantages of combining uses and future MU development (e.g., extension of existing MUs or establishment of new MUs) were also analysed. The desktop study not only examined the existing MU situation in each EU coastal country, but also resulted in an initial identification of the most important factors influencing their formation.

For each of the MUs identified drivers and barriers (also added values and impacts of MU) have been identified by the MUSES consortium/partners based on their expert knowledge and desk research. (Drivers = factors promoting MU; and Barriers = factors hindering MU) [35]. Identified drivers and barriers were categorised and compiled into a joint catalogue. The drivers and barriers for MU development identified above were evaluated by applying a scoring system in a set of interviews conducted with stakeholders in each marine EU country. Drivers and barriers were divided into several categories: policy/legal/institutional, social and economic, environmental, technological. Stakeholders

were asked to define additional (to those identified by MUSES research team) drivers and barriers. Drivers and barriers were then scored by stakeholders according to their knowledge on ascale from 0 to 3 for drivers and 0 to −3 for barriers. The following categories have been applied: 0-non existing or not relevant, 1-low priority/obstacle, 2-medium priority/obstacle, 3-high priority/obstacle. If the interviewed stakeholders were not familiar with the MU concept and their knowledge was insufficient to valuate drivers or barriers, the score was given by the project partner based on reflections of the stakeholder opinions and the project partner's own expert knowledge.

Stakeholders scored drivers and barriers for the selected MUs. Their answers were combined and the averages for each driver and barrier were calculated. On this basis, for each MU, its potential was evaluated by summing up the averaged drivers' score and the averaged barriers' score. The promising MU were identified on this basis for each country, i.e., those for which the scores for promoting factors (drivers) prevailed over obstructing ones (barriers).

Thus, these stakeholder scores provided an overall understanding of the importance of drivers and barriers for the MUs and their potential.

The drivers and barriers that affect MUs were subsequently examined during the course of 195 stakeholder interviews and three face to face workshops with 76 participants altogether. Interviews were either conducted personally or via emails or phone calls. Of these, 37 stakeholders in the Eastern Atlantic (from five countries including UK) provided their opinion. Also, 38 stakeholders were interviewed from the North Sea basin (also five countries including the U.K.). In the Baltic Sea region, a total number of 48 stakeholders were interviewed from 8 countries. The panel of interviewed stakeholders for the Mediterranean Sea consisted of 53 respondents from 6 countries. In the Black Sea 19 stakeholders were interviewed from the two EU countries. Figure 1 presents the professional composition of the stakeholders interviewed.

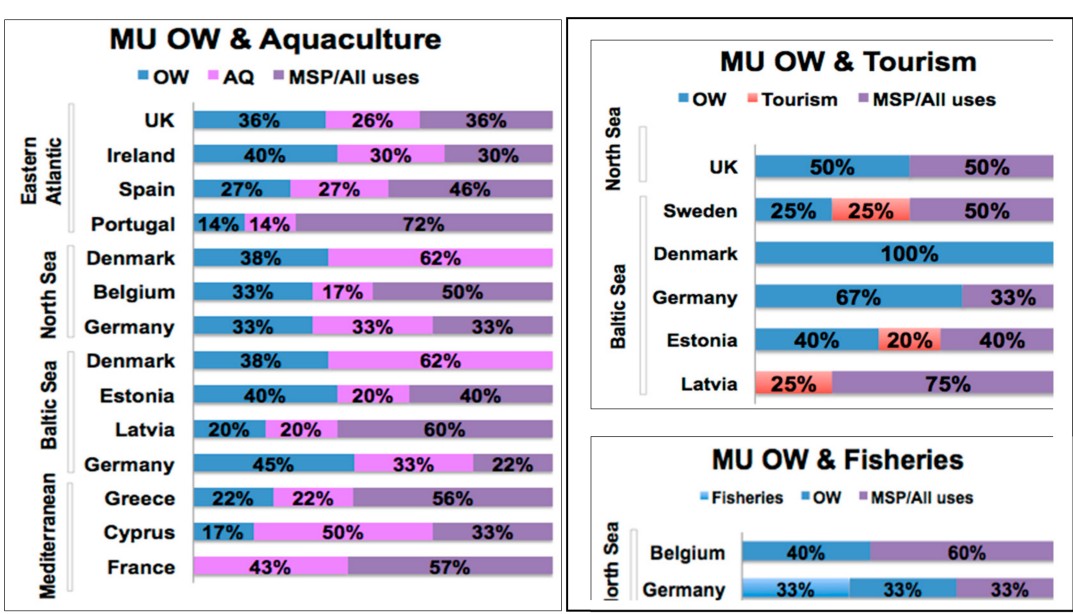

**Figure 1.** *Cont.*

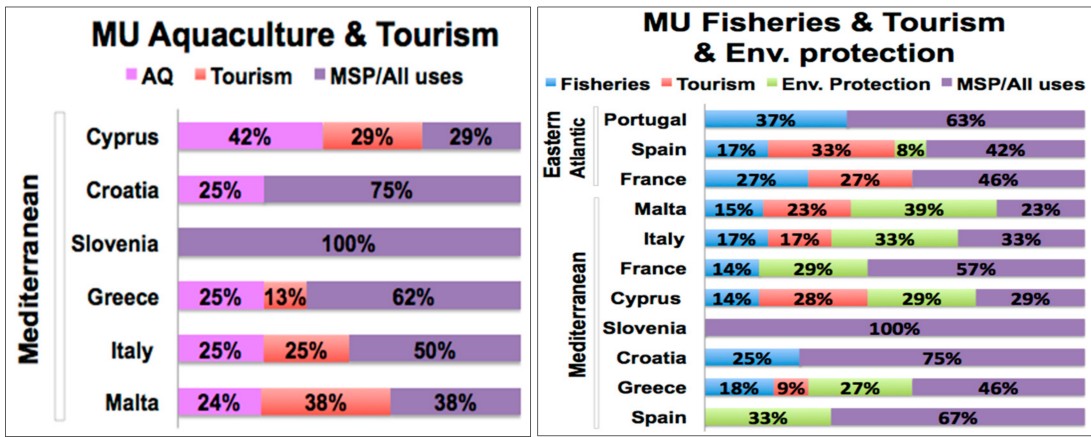

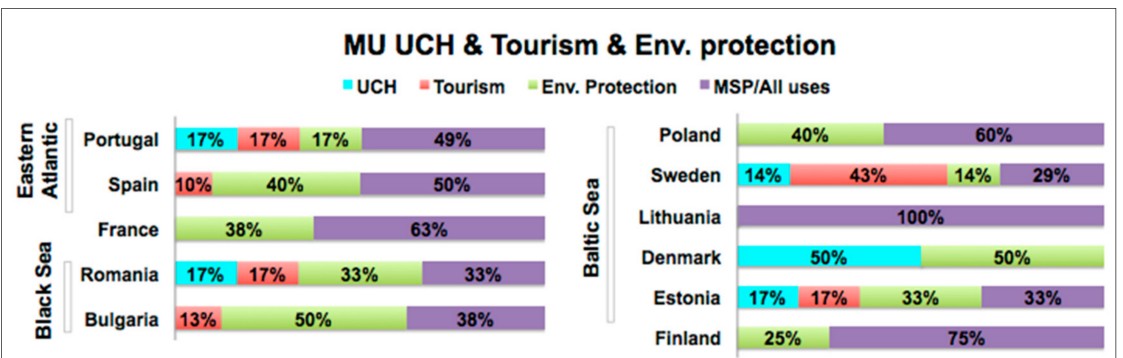

**Figure 1.** Share of stakeholders interviewed in the respective MU categories. Source: own elaboration on the basis of MUSES data.

Explanations: OW—Offshore Wind Energy (offshore wind mill parks), UCH—Underwater Cultural Heritage (e.g., wrecks, ancient constructions, paleo-landscapes etc.)

The snowball method was used for selecting the stakeholders: Those more active in the promotion of MU pointed out others that might be important for MU development in a given country. The reason for employing this method is its ability to reveal tacit knowledge of organic social networks and provide access to 'hidden populations' [41]. In our case those communities working on MSP and MU, since both of them belong to wicked non-standard problems. For such communities it is difficult to identify their size and boundaries and therefore random sampling cannot be applied [42]. Such referral sampling is also cost-efficient and provides greater rate of responses [43]. However, this method suffers from some disadvantages. Probability of selection is unknown, and thus traditional statistical methods of analyzing data are excluded [44]. However, the main risk is the risk of un-representative results due to over-representation of some groups (members of a given social network) and underrepresentation of the other stakeholders [43]. To minimise such a bias purposeful sampling was also employed in parallel. In each EU coastal country, key regulators and researchers responsible for, marine governance were approached. Interviewed stakeholders, representing a mix of sectors, regulators, NGOs, and researchers, expressed their opinions on MU in qualitative and quantitative (scoring) terms. Workshops served mainly discussing of the preliminary research findings with the stakeholders.

## 4. Results

### 4.1. The Existence of MUs

Despite several encouraging policies at national level, in the examined EU sea basins (see next chapter), MUs were at an early stage of development, primarily in the trial and pilot phase (Table 1). Existing full-fledge MUs (Table 1 blue colour) mainly concern

aquaculture and environmental protection combined with fishing, tourism, and underwater cultural heritage. These MUs are predominantly found in the Atlantic and North Sea countries.

**Table 1.** MUs in five EU sea basins. Source: [36].

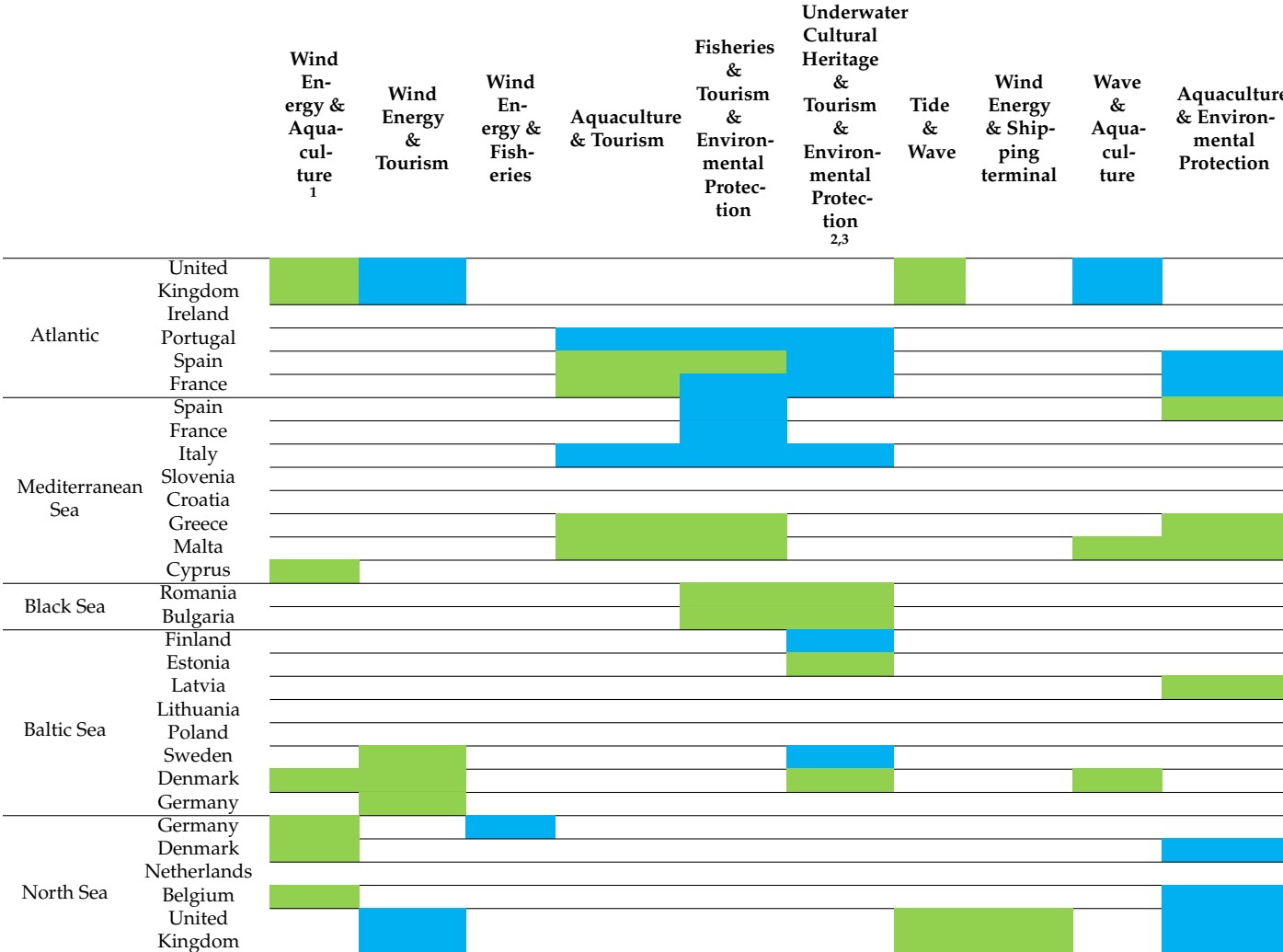

[1] In case of Cyprus (Med) WIND ENERGY devices were tested. [2] In case of Estonia this MU involves only Tourism & Environmental Protection. [3] In case of Black Sea this MU involves only UCH &Tourism. Existing and past pilot/test trials in the real environment—green. Ongoing MU in the real environment—blue.

In the North Sea countries, offshore wind energy has been combined with fishery, tourism, aquaculture and shipping terminals, whereas aquaculture has been combined with environmental protection and wave energy, in addition to wind energy. The UK has demonstrated some experience in the combination of wave and tidal energy, specifically in the Northern part of Scotland [13], as well as in the combination of tourism with offshore wind energy (The Scroby Sands Visitor Centre (UK) attracts over 35,000 visitors each year and visits to the offshore wind farm in Great Yarmouth). Testing for offshore wave energy generation and mussel aquaculture has been conducted at the Danish Wave Energy Test Centre. In addition, in Denmark, the combination of environmental protection and aquaculture has assumed the form of 'sea gardens', i.e., sites for the production of shellfish and seaweed located within NATURA 2000 sites.

In the Atlantic countries, the development of MU is uneven [34]. In the northern part, the dominance of the energy sector seems to be a driver of MU. In the Atlantic area around the UK, shellfish aquaculture trials were performed in the North Hoyle

OW farm. Furthermore, aquaculture has been combined with wave energy in Mingary Bay, West Scotland as a pilot program. For Portugal, Spain and France, the majority of existing or tested combinations involve fisheries, tourism, environmental protection, and aquaculture. For instance, in Spain, pescatourism is enhanced by law, allowing the diversification of fishing and aquaculture activities through tourism. In all three countries, such a combination is popular in various marine sanctuaries and provides the local population with a source of additional income that might have been lost due to environmental protection.

In the Mediterranean Sea, several tentative MUs were identified in five out of eight Mediterranean countries, specifically with regard to fisheries, and in three countries with regard to the combination of tourism with nature conservation. This translates into pescatourism (e.g., Italy, France, Greece), leisure boating, and marine recreation (e.g., France, Greece). Projects combining aquaculture with tourism have been identified in three countries; Malta's focus regarding MU is clearly on aquaculture.

In the western part of the Baltic Sea, the leading role in MU is assumed by wind energy in combination with tourism and aquaculture. Here, tourism related to wind farms is organised mainly by the energy sector, even including diving around the turbine foundations. In the eastern and western part of the Baltic Sea, environmental protection sites and underwater cultural heritage sites are used for tourism. Here, tourism plays the role of a catalyst. For instance, in Finland, the Kymenlaakso spatial plan contains planning solutions that seek to promote the nature tourism in combination with protection of cultural heritage.

In the Black Sea, environmental protection is a key driver for MUs. So far, it has been combined with tourism (the second important driving sector), fisheries and underwater cultural heritage. MUs have approached or become part of marine protected areas, such as the Bulgarian "Complex Kaliakra" Romanian "Vama Veche—May 2 marine reserve".

To sum up, the sea basin review reveals that the existing MUs are still largely in the pilot or conceptual phase, particularly those MUs that need policy support. This may be due to a lack of drivers, or severe barriers to MU policy support, such as policies that hamper the development of MU or do not offer the necessary support. The key finding from the comparative analysis of sea basins is that, in order to be successful, two out of three players must have the aim of achieving MU, which can either be two distinct sectors, like offshore wind and aquaculture, or a sector and a regulator. The second option prevails in practice, although two sectors were also found to initiate MU if the co-operation is driven by micro-economic benefits to both parties (such as in the case of pescatourism). This popular initiative in Southern Europe has developed as a private initiative of the fisheries sector and was only marginally facilitated by legal arrangements and financial incentives [45].

### 4.2. MUs in the EU Sea Basins from the Perspective of Stakeholders

Applying the methodology described in Section 3, six MUs were (Table 2 red colour) indicated by the stakeholders as the most relevant and with the highest potential for future development, at least in some of the countries in the five sea basins analysed. In the opinion of stakeholders, the North Sea holds the greatest potential for MUs that require the long-term installation of major infrastructure (e.g., pillars, platforms, cables, pipelines) and the development of technologies. In the Mediterranean and Black Sea, the highest potential rests with mobile and floating uses, or those uses related to the natural environment, which often require less investment and no large-scale infrastructure or development of new technologies. The Baltic Sea is divided on this issue, in that MU opportunities in the western areas are perceived as similar to the North Sea while the eastern areas consider the MU types prevailing in the Mediterranean and Black seas as more appropriate. A similar divide between the North and South can be observed in the Atlantic sea basin.

**Table 2.** Most relevant MUs derived from five EU seas. Source: [36,46].

| | MU Name | Atlantic | North Sea | Baltic Sea | Mediterranean | Black Sea |
|---|---|---|---|---|---|---|
| MU1 | Offshore Wind and Aquaculture | 1/2 | 3/1 | 1/3 | 1/1 | - |
| MU2 | Offshore Wind and Tourism | 1/1 | 1 | 3/2 | - | - |
| MU3 | Offshore Wind and Fisheries | 1 | 4 | 1 | - | - |
| MU4 | Aquaculture and Tourism | 3/1 | - | 1 | 3/3 | 2 |
| MU5 | Fisheries and Tourism and Environmental Protection | 3 | - | 1/1 | 5/3 | 2 |
| MU6 | Underwater Cultural Heritage and Tourism and Environmental Protection | 3 | - | 4/2 | 1/4 | 2 |

Explanation: The 'existing' category includes trial/pilot cases in the real environment that are ongoing or have been discontinued. The 'having potential' category constitutes hypothetical MU combinations considered by stakeholders as the most promising.

Note: the colour green indicates the number of countries within the sea basin in which the given MU exists, red indicates the number of countries in which the given MU has potential, as one use is already in place.

Positive scoring of the potential means that stakeholders scored drivers higher than barriers. For instance, in the Baltic Sea drivers for Offshore Wind and Aquaculture prevailed over barriers in the case of three countries: Germany, Latvia, and Sweden. Stakeholders scored high on non-existent MU, i.e., off-shore wind farms and fisheries; however, it is important to note that, in recently adopted maritime spatial plans, this particular combination of uses has been promoted. Even in Germany, where fishery in wind farms is prohibited, this approach is expected to change soon.

## 5. Discussion

### 5.1. Stakeholders Perception of MU Potential in Five EU Sea Basins

Table 2 presents both the current status of MUs and their development potential, highlighting the different patterns of MU contribution to sustainable maritime development in various EU sea basins.

In the North Sea, Western Baltic, and North part of the Atlantic ocean, the existing and promising MU, driven by the development of offshore energy, may deliver significant environmental benefits (green energy) as well as economic growth, potentially manifested as the development of new industries, the creation of jobs, and technological advancement. Yet, these benefits are single-use products, and a combination with other uses may serve to simply improve public perception of this new sea intruder. Nevertheless, if the combination of offshore energy with other uses is evaluated from the multi-use perspective, it does offer an improved range of costs and benefits relative to its separate use development. For instance, it creates new opportunities for the more traditional local industries, such as tourism, and may reduce the extent of spatial exclusion of other industries, such as fishery. However, several important concerns and uncertainties may arise in this case. A key problem relates to cumulative environmental impacts and social sustainability. The high concentration of human marine activity might exceed the weight capacity of the marine environment. Unfortunately, the scientific evidence regarding this issue is scarce; thus, the territorial governance is bound to learn by doing. More evidence regarding social sustainability in the case of German or Polish MSP has been collected. Assuming the fair participation of all stakeholders in the governance processes, and in MSP in particular, does not eliminate the main issue of well-organised stakeholders, such as offshore farm developers or navigation, dominating the maritime governance processes. This may result in an inadequate recognition of some stakes, such as is the case with the abovementioned emotional bond people feel toward the sea or a neglecting of local culture and tradition. Moreover, the distribution of positives and negatives may occur at the expense of the weaker social groups, who are unable to properly communicate their fears. Parts of these problems, such as the prevention of landscape pollution and the preference of offshore farms open for local uses, can be addressed by MSP. However, MSP must be effectively combined with other governance processes, such as stakeholder capacity building, the

education of the general public, as well as of offshore developers on the consequences of their dominance, and the lowering of transaction costs related to the social dialogue on MU development for the Eastern Baltic Sea.

The MU development of the Mediterranean and Black seas creates less of a threat for local cultures, traditions, and marine industries. The economic benefits are also evident for stakeholders, who are motivated to undertake various MUs, such as by combining cultural heritage, tourism, fisheries, and natural endowments. The MARSPLAN-BS II Black Sea project (2019–2021) has recently elaborated case study on MU of Tourism, UCH, and Environmental protection (in particular for Bulgaria) and how the identified barriers could be addressed with the MSP. Interviewed citizens positively assess economic aspects of the MUA, but the key concern remains the increase in the exploitation of the natural/cultural capital of the sea and oceans—in particular, risks of devastation of cultural heritage. Here, the role of MSP is limited. Prohibited areas may be effective in reducing the pressure but perhaps more important is the introduction of coherent rules and the regulated intensity of such MUs, the education of all stakeholders involved on the risks of excessive marine resource exploitation, as well as the monitoring and reaction system to the environmental damages that may arise.

Some of these problems have been recognised by stakeholders as barriers to MU development. For instance, in the case of a combination of the three sectors of fisheries, tourism, and environmental protection, the stakeholders underlined the resistance to change and limited expertise among small fishing communities, as well as the lack of new ideas regarding the organised economic businesses of fishers. In relation to the combination of the underwater cultural heritage (UCH), tourism, and environmental protection sectors, the stakeholders pinpointed the risk of looting, the deterioration and destruction of UCH sites and, with regard to the combination of offshore wind and aquaculture, the resistance of civil society and fishers to offshore wind farms was highlighted [34].

However, the consistently highly scored barriers in the study related to both administrative procedures and technological gaps, as well as also to the lack of political will to enhance MU, an insufficient will to cooperate among respective sectors, and narrow policy design. The most important barriers revealed by stakeholders are related to the "silos" structure of policy making and related rules and procedures. Thus, one should advise that MU friendly legislative frameworks and administrative procedures (e.g., joint permits or environmental reports for each of the combined uses) should be installed at national level. They can come as a result of evaluation of maritime spatial plans from the perspective of MU enhancement.

Stakeholders were convinced that the market alone might be insufficient to drive combined development for many types of MUs and they considered existing public support insufficient. One can interpret that MUs cannot be supported as it is done now i.e., mainly by maritime spatial planning (requesting MUs, separating space for MUs) and R&D (financing development of technology or other research on MU).

*5.2. Supporting MU*

According to stakeholders' perceptions, policy makers (relevant ministries) are the most important national actors to promote MU development and boost communication by spreading information and bringing together sectors relevant for the MU combinations. In practice, however, the MU concept is supported mainly by maritime spatial planning and through national sectoral policies, but, with few exceptions, it is not included within the broader blue growth set-up or any environmental strategies. This created a policy mismatch. In the paper entitled "Blue Growth opportunities for marine and maritime sustainable growth" [47] no mention is made of multi-use, co-location, co-existence, or co-use. The only indirect mention is where the Commission praises countries for locating cages "along with offshore wind farms" [47] in order to avoid spatial conflicts. However, the EU supports MU as a research topic (mainly R&D) through 7FP and Horizon 2020. Other EU policies recognize the potential of MU as an efficient use of resources only in a

few cases. In particular, the economic incentives from the European Maritime and Fishery Fund, bringing together Fisheries Local Action Groups (FLAGs), is a major instrument toward the development of pilot MU projects. The specific policy component of MU related to resource-sharing makes it an important instrument of circular economy, garnering increasing attention and support from the EU Commission but MU is still absent from the EU Marine Framework Directive and the 14th Sustainable Development Goal of the UN (MU is implicitly present in Goal 12 since MUmight offer "fewer resources per unit of production").

Support at Macroregional level seems uneven.

- In the "Sustainable Blue Growth Agenda for the Baltic Sea Region" of 2014 there is a suggestion of supporting flagship projects related to exploiting the potential for co-existence of maritime uses [48].
- The Black Sea a New Regional Cooperation Initiative named Synergy [49] contains no single reference to MU, collocation, co-existence nor co-use. However, this initiative is not limited to blue growth but encompasses broader array of co-operation topics. Also, Common Maritime Agenda (Ministerial Declaration since 2019) and SRIA (Strategic Research and Innovation Agenda) for the Black Sea Basin (which replace the Synergy from 2007) do not include any reference to Maritime Spatial Planning and MU (although the MSP process for the Black Sea Basin (Bulgaria and Romania) was started in 2014 under the two pilot projects MSRSPLAN-BS I and II, supporting the ongoing process of MSP in both EU Member States).
- The Ministerial Declarations of the Union for the Mediterranean on the blue economy [50] also contains no links to the MU concept. However, in the EU Strategy for the Adriatic and Ionian Region [51,52] some specific claims towards the development of the MU approach can be found. Coordination of aquaculture and fisheries with other activities (tourism, environmental protection) is suggested although it is not clear whether the final outcome should be in MU form or co-existence. Also, the Strategic Research and Innovation Agenda of the Blue Med Initiative [53] explicitly address MU (multi-use platforms in support of environmental monitoring, safety and security, and renewable energy development). In the Declaration of the Meeting of the Ministers of the Countries participating in the Initiative for the Sustainable Development of the Blue Economy in the Western Mediterranean the ministers support multi-use offshore platforms and ask MSP for synergic uses of sea space and resources [54].
- In the Atlantic, the Atlantic Strategy was considered as a key blue growth document. The Action Plan for the Atlantic Strategy contains suggestions on integration of renewable energy installations for offshore wind, wave, tidal, and biomass energies with desalination plants and multipurpose offshore platforms [55].
- In the North Sea the blue growth strategy is under elaboration within preparatory action' for a regional strategy in the North Sea region supporting regional cross-sectoral maritime cooperation. At the workshop on Strategic Cooperation on Blue Growth in the North Sea the MU have been discussed in depth, in particular in relation to technologies that ensure the multi-use of maritime space [56].

The recent blue growth documents possess more links to the MU concept. However, they address rather singular topics important in the context of a given sea basin (e.g., multi-purpose platforms for the Atlantic Region or multi-use technologies for the North Sea region) rather than MU as a systemic approach. Therefore, the support is very general if any. One can rather observe mismatch of terms and notions. Therefore, support is directed towards combination of different uses without specifying any details.

A similar situation has been noticed at the national level. It was discussed in detail by Przedrzymirska et al. [36,46], and therefore in this paper only the most important results are referred to.

1. The strongest policy support is provided by the U.K. and by some Mediterranean countries. The latter enhance sea uses combination in various types of policy doc-

uments, support schemes and national legislation [57] but only in a few sectors (pescatourism as a leader). In the U.K. support is mainly provided by MSP. It is expressed in the legal/policy MSP documents (e.g., in the Marine Policy Statement [58] that explicitly states that Marine Plans could 'encourage co-existence of multiple uses'). This co-existence focus is generally observed in the Marine Plans for England, Scotland, and Northern Ireland.

2.  MU has been supported at national level in the Black Sea region. However, this concept is absent in the strategic documents at the regional level (sea-basin level) with exception of EU funded projects (MU concept and its support with MSP has been elaborated in the MARSPLAN-BS II project).

3.  In the North Sea support exists at various forms (legislation, economic incentives, administrative routines, MSP, sectoral strategies) in almost all countries, with the exception of Denmark in which economic incentives and MSP enhancing MU are missing. In the North Sea the various terms supporting co-combination of uses are used in the national legislation of the majority of sea basin countries, but their practical usage is limited to the UK and Belgium. However, in contrast to the Mediterranean case, maritime spatial planning in the North Sea countries is much more open to MU in the planning stage. However, the support is also very general.

4.  In the Baltic Sea Region MU support is limited to the western countries including Poland. However, this support is hardly systematic. For instance, in Poland MU is supported mainly by MSP. Economic incentives for MU are missing despite verbal support to this concept in regional strategies. MU is also missing in the national legislation of majority of countries in the Baltic Sea Region. MU is not excluded there but is not directly supported.

The entire picture is dynamic. For instance, French National Strategy for the Sea and Coast [59] includes a set of priority actions including stimulation of the blue economy and innovation, the development of synergies among existing and novel uses of the sea, and preservation and sustainable use of the marine environment and its resources. The strategy states the infrastructure sector must be interested in the prospects opened up by multi-purpose offshore platforms. These would enable development of zones of activities at sea, facilitating the establishment of facilities for the development of maritime resources.

### 5.3. Contribution of MU to Sustainable Development

This research revealed several crucial problems of MU development as an instrument for sustainable spatial (maritime) governance, which require conscious efforts at various geographical scales. The solutions to these problems are presented below.

### 5.3.1. Need for the Territorialisation of MUs

MUs contribute in different ways to the sustainable maritime development in various sea basins. Thus, if one has been determined to support this use of maritime space, different approaches must be applied in different sea basins in order to mitigate any potential negative effects of MUs. For the Eastern Baltic Sea, the Mediterranean Sea, and the Black Sea, the focus should remain on the environmental and cultural pillar of sustainable development. For this purpose, an existing co-operation framework can be used, such as the Helsinki Commission in the Baltic Sea, the Barcelona Convention in the Mediterranean, or the Black Sea Commission in the Black Sea. These are the proper fora to discuss the consequences of MU enhancement at the sea basin level. In the North Sea, North Atlantic, and Western Baltic Sea, a rise in the public's awareness regarding MSP and other governance processes seem to be the two most important preconditions for the implementation of MUs. The latter can be achieved in the framework of transnational projects initiated by DG Mare at each sea basin. Efforts must be put in place that help to alleviate the potential consequences of a maritime governance that is dominated by strong economic sectors (e.g., spatial disorder at sea).

### 5.3.2. Global Recognition and Agency-Driven, Content-Oriented Coordination

The sustainable aspect of MU is underrepresented in policy documents, at least at the EU and sea basin level. This may be due to research gaps in the evaluation of the broader socio-economic benefits of MU (see next conclusion) as well as to a limited understanding of MU, as a concept, among the general public and regulators. EU macroregional strategies and other sea basin frameworks (e.g., Interreg programmes) should be more actively used in order to alter the situation and to integrate (better link) various MU implications for blue growth, MSFD, and social sustainability. The same applies to SDGs goals in particular the 14th one. The potential of the sea basin in terms of exchanging experience and encouraging the exploration of new concepts and ideas should be better exploited in the case of MU. Such a debate is necessary, not only for MU enhancement but also for the recognition of its pros and cons regarding maritime sustainable development. The divides within sea basins on the types of MUs with the highest potential may also call for a bilateral, agency-driven, content-oriented coordination [60] between countries or even regions managing marine space. In these cases, sea basin collaboration may not be sufficient.

### 5.3.3. More Holistic MU Support

Public support for MU requires a much more complex approach. Economic incentives should cover high transaction costs if they are prohibitive. This cannot be done under maritime spatial planning. Some other incentives are necessary if private costs of MUs exceeds the private benefits but considerable positive externalities do exist. This must be done in the form of various subsidies, but the problem is with monetisation of such externalities. Policy systems are not prepared to take on such tasks. The holistic approach, i.e., complementing R&D and maritime spatial planning with financial incentives for private sector and educational efforts, is crucial. A precondition is a firm policy commitment at various levels and vertical and horizontal policy coordination in order to avoid contradictory policy incentives.

### 6. Conclusions

The above presented research has identified the following conditions of turning MU into effective instrument supporting sustainable development of marine space:

- Recognition of specifies of different EU sea basin in terms of impact of MU on various aspects of sustainable development
- Recognition of MU not only in MSP documents but also in other key strategies such as SDGs goals or Macroregional strategies and action plans
- Employment of more coherent and concise spectrum of MU support efforts covering not only MSP but also sectoral incentives and educational efforts.

To achieve these goals there is a need for more profound research on MU. The following research topics are relevant in terms of enhancing MU as a sustainable development component.

- Evaluation of policy performance with regard to supporting MUs. In particular, more research is necessary on how to integrate an MU concept within the blue growth or environment policy and social sustainability idea. While the initial analysis indicated that the MU concept is well aligned with sustainable development ambitions, reality does not support this finding. Although MU fits into the sustainable development in real applications in maritime governance it is decidedly absent.
- More profound research on MUs, and the impact MU has on societal goals. Typically, such goals include enhancing societal innovations, securing places for less influential but socially significant sectors, leaving more space for the decisions of future generations, mitigating climate change, contributing to the well-being of the environment, etc. However, the focus should remain on recognition and distribution. MUs might create several risks in this regard that require further investigations.

- Evaluation and assessment of various types of costs and benefits, including external ones related to MUs and their incorporation into the price. The outcomes of such research can guide both the private sector's decisions and the decision-makers' allocation of sea space for various uses.
- Compiling spatio-socio-economic multipliers of MUs (e.g., impact of MUs on spatial patterns, local cultures and economic growth on land).

**Author Contributions:** Conceptualization: J.P., I.L., J.Z., H.C., A.B., D.D. and K.S.; Methodology: D.D., J.P., M.B.; Validation: C.O., K.G., K.S., M.M. (Monika Michałek), M.C.V., K.G.-H. Formal Analysis, Investigation, Resources, and Data Curation: J.P., M.B., M.C.V., A.B., M.d.S.V., A.S.-Z., V.O., E.P., G.K., M.F.S., A.B., M.L., R.L., I.L., M.M. (Mairi Maniopoulou), V.V., Z.K., C.O.; Writing—Original Draft Preparation: J.P., I.L., J.Z., H.C., B.H.B., E.R. and D.D.; Writing—Review & Editing, J.Z., K.G., M.M. (Monika Michałek), B.H.B.; Visualization: M.L., I.L., M.C.V., S.S.; Supervision: J.P., J.Z.; Project Administration: J.P.; Funding Acquisition: B.B., A.S.-Z., J.P. All authors have read and agreed to the published version of the manuscript.

**Funding:** Paper presents research outcomes of the MUSES project, which has received funding from the European Union's Horizon 2020 research and innovation programme under grant agreement no. 727451. The parts relating MUSES findings to maritime sustainable development were prepared with support of the project "Spatio-economic multiplier in maritime economy" financed by the Polish National Science Centre under the grant agreement UMO-2018/31/B/HS4/03890.

**Institutional Review Board Statement:** Not applicable.

**Informed Consent Statement:** Informed consent procedure was applied to all subjects involved in the study.

**Data Availability Statement:** The individual data underlying this article (interviewees' responses) cannot be shared publicly due to the privacy of individuals that participated in the interviews. The aggregated data at the sea basin or national level ("joint sheets") can be shared on request to the corresponding author.

**Acknowledgments:** We are very grateful for useful insight and initiation of preparation of the paper to all colleagues from the MUSES project. We express gratitude to Tomasz Laskowicz for preparation this paper to uploading in a required format. We are thankful to all MUSES stakeholders for sharing their knowledge with us.

**Conflicts of Interest:** The authors declare no conflict of interest.

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
