# Peer review of "Multi-Use of the Sea as a Sustainable Development Instrument in Five EU Sea Basins"

_sustainability, doi:10.3390/su13158159_

Round 1

Reviewer 1 Report

The manuscript entitled “Multi-Use of the Sea as a Sustainable Development Instrument in Five EU Sea Basins” addresses an important topic of exploring the potential of multi-use of the ocean and marine spaces and resources in five EU sea basins, towards more sustainable model of blue economy and other social and environmental benefits. The paper is very well written, with well-proportional and logical structure. Data and methods are clear and correct, well described and explaining the applied methodology. All sections in the paper are additionally supported and visualized by applied figures and tables and graphical abstract. I have only few observations and specific comments/suggestions that I believe will help improve the paper quality:

  1. The abstract looks oversimplified and should include briefly main results and conclusions of the research.
  2. Introduction: although the paper has clearly stated research objectives grouped in two main aims, I recommend to point what is the added value of the research and this should be explicitly emphasized at the end of introduction part.
  3. Materials and methods: a little more details need to be given to the stakeholder engagement, as the MUSES results were based on active stakeholder participation.
  4. Discussion part, line 325: this statement is no longer correct - the MARSPLAN-BS II Black Sea project (2019-2021) has recently elaborated case study on MU of Tourism, UCH and Environmental protection (in particular for Bulgaria) and how the identified barriers could be addressed with the MSP (http://www.marsplan.ro/en/results/marsplan-bs-ii-addressing-the-multi-use-concept.html). MU concept has been also referred in the national MSP of Bulgaria and Romania.
  5. Discussion part, lines-373-374: there is also Common Maritime Agenda (Ministerial Declaration since 2019) and SRIA (Strategic Research and Innovation Agenda) for the Black Sea Basin (which replace the Synergy from 2007), however both documents do not include any reference to Maritime Spatial Planning or to MU concept or co-location, co-use, synergies etc. (although the MSP process for the Black Sea Basin (Bulgaria and Romania) has been started since 2014 under the two pilot projects MSRSPLAN-BS I and II, supporting the ongoing process of MSP in both EU Member States.
  6. Discussion, lines 420-421: this statement is no longer correct - as pointed above recent study on MU concept and its support with MSP has been elaborated in the MARSPLAN-BS II project, as well as the MU is included in the national MSPs of Bulgaria and Romania. Thus, the MU concept is already supported at national and regional level. 

Reviewer 2 Report

Comments to the Author

Dear authors,

Please find below my review report.

Thank you for submitting your manuscript. This manuscript has potential to publish, however, a revisit is needed before considering for publication. Please see below the comments. In general, it is a well-written paper with a clear outline, abundant literature review, and sufficient discussions.

In this paper, the authors fully discussed the concept, the potentials and the problems of maritime multi-use through the analysis of the current situation in the Five EU Sea Basins.

Please see below the comments.

p.5; line 184: In Figure 1 acronyms ‘OW’ and ‘UCH’ appear for the first time and there is no explanation of what they mean. For ‘UCH’ there is an explanation much later on the page 9;line 335.

Section 3; p.5; lines 186-203: A revisit is needed. More specifically, literature on the method used (snowball method) is necessary. What are its strengths and weaknesses? Also, more details on drivers and barriers scoring are needed

p.8;line 281: In table 2 there is not orange color.

Sections 5 and 6: Revisit these two sections by reorganized Discussion in 2-3 dedicated sub-sections. I believe that the majority of the content of the Conclusions can be included in the Discussion section. It would be excellent to make a concise conclusion. Please shorten it.

p.15; line 587: Please fix reference 19. The correct is ‘Tsilimigkas, G. and Rempis,N. (2017)…. https://doi.org/10.1016/j.ocecoaman.2017.02.001.
